# Salvage Surgery for Small-Cell Lung Cancer—A Literature Review

**DOI:** 10.3390/cancers15082241

**Published:** 2023-04-11

**Authors:** Natalia Motas, Veronica Manolache, Marco Scarci, Victor Nimigean, Vanda Roxana Nimigean, Laurentiu Simion, Madalina Cristiana Mizea, Oana Gabriela Trifanescu, Bianca Galateanu, Mirela Gherghe, Cristina Mirela Capsa, Diego Gonzalez-Rivas, Mihnea Dan Davidescu

**Affiliations:** 1Discipline of Thoracic Surgery, “Carol Davila” University of Medicine and Pharmacy, 022328 Bucharest, Romania; 2Department of Thoracic Surgery, “Prof. Dr. Al. Trestioreanu” Institute of Oncology, 022328 Bucharest, Romania; 3Department of Thoracic Surgery, Oncology Hospital Memorial, 013812 Bucharest, Romania; 4Department of Thoracic Surgery, Imperial College NHS Healthcare, Hammersmith Hospital Du Cane Road, London W12 0HS, UK; 5Department of Anatomy, Faculty of Dentistry, “Carol Davila” University of Medicine and Pharmacy in Bucharest, 022328 Bucharest, Romania; 6Department of Oral Rehabilitation, Faculty of Dentistry, “Carol Davila” University of Medicine and Pharmacy in Bucharest, 022328 Bucharest, Romania; 7Discipline of General and Oncological Surgery, “Carol Davila” University of Medicine and Pharmacy, 022328 Bucharest, Romania; 8Department of General and Oncologic Surgery, “Prof. Dr. Al. Trestioreanu” Institute of Oncology, 022328 Bucharest, Romania; 9Discipline of Oncology, “Carol Davila” University of Medicine and Pharmacy, 022328 Bucharest, Romania; 10Department of Radiotherapy, “Prof. Dr. Al. Trestioreanu” Institute of Oncology, 022328 Bucharest, Romania; 11Department of Biochemistry and Molecular Biology, University of Bucharest, 022328 Bucharest, Romania; 12Discipline of Nuclear Medicine, “Prof. Dr. Al. Trestioreanu” Institute of Oncology and University Emergency Military Hospital “Dr. Carol Davila”, 022328 Bucharest, Romania; 13Department of Radiology “Prof. Dr. Alexandru Trestioreanu”, 022328 Bucharest, Romania; 14Department of Thoracic Surgery and Minimally Invasive Thoracic Surgery Unit (UCTMI), Coruña University Hospital, 15008 Coruña, Spain

**Keywords:** lung cancer, small-cell lung cancer (SCLC), salvage surgery, salvation surgery, salvage resection, lung resection, definitive chemoradiotherapy, locally advanced lung cancer, definitive surgery, resection after chemoradiation

## Abstract

**Simple Summary:**

Salvage lung resection for lung cancer is a relatively new concept, and most of the operated patients present non-small cell lung cancer (NSCLC). For small-cell lung cancer (SCLC), salvage surgery is extremely rarely performed, in highly selected patients proved with limited disease after radical chemoradiation, as an alternative to second-line chemotherapy. From the published cases, it appeared that salvation surgery for SCLC presents postoperative morbidity similar to salvage surgery for NSCLC, no postoperative mortality, and a median estimated overall survival of 92% at 2 years and 66% at 5 years. Salvage lung resection for SCLC may be a reasonable treatment in very highly selected patients, offering good local control and a favorable survival outcome.

**Abstract:**

(1) Background: Salvation surgery for small-cell lung cancer (SCLC) is exceptionally performed, and only a few cases are published. (2) Methods: There are 6 publications that present 17 cases of salvation surgery for SCLC—the salvation surgery was performed in the context of modern clearly established protocols for SCLC and after including SCLC in the TNM (tumor, node, metastasis) staging in 2010. (3) Results: After a median follow-up of 29 months, the estimated overall survival (OS) was 86 months. The median estimated 2-year survival was 92%, and the median estimated 5-year survival was 66%. (4) Conclusion: Salvage surgery for SCLC is a relatively new and extremely uncommon concept and represents an alternative to second-line chemotherapy. It is valuable because it may offer a reasonable treatment for selected patients, good local control, and a favorable survival outcome.

## 1. Introduction

Small-cell lung cancer (SCLC) represents approximately 15% of all lung cancers. SCLC is characterized by rapid growth and early lymphatic and hematogenous metastases. For this reason, surgical resection is not the first therapeutic option; in fact, surgery is not an option for most patients with SCLC. The prognosis without treatment is very poor: a few months of survival. Initially, SCLC responds very well to chemotherapy and radiotherapy; however, resistance occurs quickly, and recurrent disease is the usual cause of death. Classically, it is stated that patients with treated limited-stage disease present about 15% survival at 5 years, and patients treated for extensive-stage disease have less than 5% survival at 2 years [1,2]. The current 8th and the previous 7th edition of the TNM classification for lung cancer include SCLC in pathological TNM staging based on survival analyses performed for resected patients with SCLC [3]. Shields et collab. were the first to suggest in 1982 the importance of TNM staging in SCLC, also based on the survival benefit of surgery in early-stage SCLC [4].

Salvage surgery is a relatively new entity in thoracic surgery and oncology. In the context of modern standard therapies for SCLC, we found the first indications for salvage surgery in SCLC in a publication from 2006 (Anraku and Waddell), which were identified as chemo-resistant localized SCLC or local relapse after an initial response to chemo/chemoradiotherapy, for the reason that resection might be more effective than second-line chemotherapy [5].

## 2. Materials and Methods

Search strategy and selection criteria: references for this review were identified through searches of PubMed with the search terms “salvage” and/or “salvation” and “surgery” or “resection” or “operation”, for the period until December 2022. The final reference list was generated based on originality and relevance to the broad scope of this Review: resections performed for SCLC, with the intent of cure, after (or during) complete chemoradiation performed with cure intent, surgery not being included in the initial management plan for each of the cases. Each case was described in detail in the published papers. Exclusion criteria were: lack of detailed information about each patient who underwent salvage surgery for SCLC, surgery performed as a part of the initial plan, lack of SCLC confirmation before any treatment, resection per primam for early-stage SCLC, cases with salvage surgery for NSCLC.

Statistical analysis was performed using SPSS version 26 for Windows. Descriptive statistics were used to characterize the patients. The oncologic outcome for SCLC patients who underwent salvage surgery was estimated using the Kaplan–Meier method to determine the median overall survival (OS).

## 3. Results

After the repeated search, we identified only 6 publications with salvage operations performed for SCLC in which specific information about every case was offered for 17 patients. We synthesized in Table 1 the data about the published 17 cases. It is important to state that all patients were confirmed with SCLC (pure SCLC or mixed/combined SCLC with an NSCLC component) before any treatment, according to each of the published papers.

The most recent publication regarding salvage surgery in SCLC is the brief report by Joosten and colleagues from 2021 [6]. The authors reported one of the largest cohorts of salvage surgery for SCLC, persistent or relapsed after chemoradiation; 10 patients were examined retrospectively, as they had been operated between 2008 and 2020 in their institution. The included patients were diagnosed with SCLC, one of them presenting a mixed form of SCLC + NSCLC. The multidisciplinary tumor board which decided the indication for surgery with salvation intent included at least “two lung cancer surgeons with varying levels of experience”. The stages before therapy were miscellaneous, and we present the detailed patient data in Table 1. Prophylactic cranial radiation was performed in 8 patients of the 10 presented, as part of the multimodality non-surgical plan specific for SCLC management. The indication for salvage surgery was local disease recurrence for nine patients and persistent local disease in one case. The time between the end of radiation therapy and the recurrence/persistence of the disease was between 3 months and 78 months. Before salvation resection, half of the patients were treated with second-line chemotherapy. Between the last day of radiation treatment and the time of resection, the time interval was of 3–80 months. All 10 patients received anatomical lung resection as salvation surgery. In total, five lobectomies, two bilobectomies, one pneumonectomy, and one segmentectomy were performed. Postoperative complications occurred in five patients (50%), but the mortality was zero at 30 days and 90 days postoperatively. In two patients, a pathological analysis found no residual tumoral tissue. The authors successfully concluded that a “multidisciplinary tumor board should consider surgical salvage for highly selected patients with locally recurrent or persistent SCLC after chemoradiotherapy” [6].One case was published by Kanayama and colleagues in 2019: a female patient treated with chemo–radiotherapy for a SCLC of the left lower lobe, with complete remission for 5 years. After 5 years, a 30 mm nodule appeared in the same lobe. The authors performed left lower lobectomy and lymph node dissection, and a pathological analysis showed combined a SCLC–large-cell carcinoma [7].Nakanishi and colleagues carried out through in 2018 an analysis of five cases of salvage surgery for SCLC after chemo–radiotherapy and reviewed the literature until that time; this excellent work was published in 2019 [8]. They found that the first papers on salvage surgery in SCLC were published in 1991 [9,10]. The results were unsatisfactory (the median survival after surgery was 13.5–18.5 months) because the studies were conducted on heterogeneous populations and before the establishment of modern standard therapies for limited-disease SCLC. The five cases presented by Nakanishi and colleagues, all male, were confirmed as limited-stage SCLC and treated with chemoradiation with curative intent. Subsequently, the patients received salvage surgery as a curative-intent lung resection for residual lesion (one case) or local re-progression in the previously irradiated area (four cases), more than 12 weeks after the last day of chemo–radiotherapy. The clinical stages before surgery were ycIA2, ycIA2, ycIA3, ycIB, and ycIB; therefore, all five patients were ycN0. As for the resection type, the authors performed three lobectomies with lymph node dissection (covering the bronchial stump with intercostal muscle flap) and two wedge resections without lymph node dissection, all through thoracotomy. Morbidity and 90-day mortality were zero. The pathological stages were ypIA2, ypIVa (pleural metastases), ypIB, ypIIB, and ypIIIA. The histology after salvage surgery showed SCLC in three cases, combined SCLC with adenocarcinoma in one case, and squamous cell carcinoma in one case. The estimated 3-year and 5-year overall survival rates were 100% and 67%, respectively [8].Another case was presented in the literature by Pan and colleagues in 2017 [11]. This was a male patient with cIIIA SCLC (cT2N2M0) not initially a candidate for surgery; due to progression under chemotherapy, he was considered chemotherapy-refractory, and salvage surgery was decided, consisting of right pneumonectomy with mediastinal lymph node dissection. No postoperative morbidity occurred. The pathological stage was pT2N2Mo—pIIIA, and the histology was SCLC. Postoperative chemotherapy and radiotherapy were administered, and the patient survived free of disease for more than 2 years [11].Eberhardt and colleagues conducted a phase II trial on a prognostically orientated multimodality approach, including surgery, for selected patients with SCLC; the results were published in 1999 [12]. Staging included CT scan (chest, upper abdomen, and brain), mediastinoscopy, and radionuclide bone scan (no PET or PET-CT). Surgery was planned for all cIB/IIA stages after four cycles of chemotherapy, all cIIB/IIIA (IIIB) stages after four cycles of chemotherapy and 45 Gy (delivered on the tumor and mediastinal nodes). Of the 46 patients included, 32 were planned for surgery and, after restaging (and excluding N2), 24 were operated. Twenty-three patients were resected, R0, but is not clear if all received chemo–radiotherapy or only chemotherapy. A pathological complete response was observed in 11 cases, SCLC histology in 9 cases, and NSCLC histology in 3 cases. The 5-year survival of those 23 patients was 63%. Relapse occurred in nine cases and only as distant metastases (no local/locoregional failure). Two patients developed a second primary cancer. Analyzing the paper, it is not clear how many patients were resected after chemoradiotherapy; the dose was only 45Gy; multimodality was given as in NSCLC; in fact, those cases do not correspond to the actual definition of salvage surgery. However, the TNM staging was probably conducted according to the 6th edition, and now those cases would be upstaged following the new definitions. We can consider that the excellent work of Eberhardt and colleagues demonstrated that surgery is feasible after chemo/chemo–radiation of SCLC in patients without N2, with good locoregional control and very good long-term survival.In 2009, Vallieres and colleagues published proposals regarding the relevance of TNM in the pathological staging of SCLC in the 7th edition of TNM—the IASLC Lung Cancer Staging Project [3]. The huge database included 12620 eligible cases of SCLC, 349 of which were completely resected R0 cases with SCLC. The number of patients in each clinical stage (6th edition of TNM staging system) was: cI, 159, cII, 71, cIIIA, 76, cIIIB, 33, and cIV, 10 patients. There were 119 patients with clinical stages III and IV. It was not specified which patients received any induction therapy and was not possible to evaluate if and how many cases corresponded to the salvage surgery definition, to serve the purpose of our study. There is no doubt that the paper is extremely valuable, and the results offered are useful for the entire medical community.

At the time of publication of each of those papers, 4 patients out of 17 (Table 1) were deceased because of the disease (patients 8, 9, 10, and 13 at 86 months, 12 months, 33 months, and 46 months, respectively) after salvage surgery. One patient was not presented in terms of survival (case 11). The rest of the 12 patients were alive and free of malignant disease.

After a median follow-up of 29 months, the estimated median overall survival (OS) was 86 months (Figure 1). We calculated the estimated median survival at 2 years as 92%, that at 3 years as 80%, and the median estimated 5-year survival as 66%.

**Table 1 cancers-15-02241-t001:** Cases of salvage surgery for SCLC identified in the literature. All patients were confirmed with an SCLC component before any treatment (according to the published papers).

Publication (in Alphabetical Order of the First Author’s Name)	Nr. crt.	Age (Years)	Gender	Stage at Diagnostic	Dose of RT (Gy)	Indication for Salvation Surgery (as Described by the Authors)	Time from RT to Surgery (mo)	Stage before Surgery (as Presented by the Authors)	Stage before Surgery cTNM	Resection Performed (as Described by Authors) **	Pathology after Surgery	Pathological Stage ypTNM	Survival (mo)
Joosten, 2021 [6]	1	71	f	cT4N2	66	Recurrence	23	r-cT1aN0	IA1	Left upper lobectomy	N/A	ypT2N1	>2
	2	58	f	cT2aN1	66	Recurrence	10	r-cT1bN0	IA2	Left upper lobectomy	N/A	ypT1cN0	>11
	3	59	f	cT3N1	56	Recurrence	23	r-cT2aN0	IB	Right upper lobectomy	N/A	ypT1cN0	>8
	4	53	f	cT4N0	N/A	Recurrence	22	r-cT4N0	IIIA	Left pneumonectomy	N/A	ypT2N0	>8
	5	64	f	cT4N2	50	Recurrence	7	r-cT1aN0	IA1	Right upper lobectomy	N/A	ypT1aN0	>34
	6	66	f	cT3N2	N/A	Recurrence	80	r-cT3N1	IIIA	Right upper bilobectomy	No malignancy	ypT0N0	>35
	7	58	m	cT3N0	66	Persistent disease	3	r-cT3N0	IIB	Right upper lobectomy	No malignancy	ypT0N0	>71
	8	58	f	cT2aN1	46	Recurrence	47	r-cT1cN0	IA3	Segmentectomy (complex) 1,2,6	N/A	ypT2aN1	86 DOD
	9	48	f	cT4N0	45	Recurrence	22	r-cT2bN1	IIB	Right upper bilobectomy	N/A	ypT2bN0	12 DOD
	10	64	m	cT1bN0	45	Recurrence	22	r-cT1cN0	IA3	Right upper lobectomy	N/A	ypT2N0	33 DOD
Kanayama, 2019 [7]	11	80	f	cT2aN0M0-cIB	45	Late progression	60 *	ycT2aN0M0-cIB	IB	Left lower lobectomy	Combined SCLC + LCLC	N/A	N/A
Nakanishi, 2019 [8]	12	59	m	cT1cN3M0-cIIIB	54	Residual lesion	18	ycT1bN0M0-ycIA2	IA2	Wedge resection	SCLC	ypT2aNxM1a-ypIVA	114 FOD
	13	61	m	cT2aN2M0-cIIIA	45	Local reprogression	10	ycT1bN0M0-ycIA2	IA2	Wedge resection	SCLC	ypT2aN0M0-ypIB	46 DOD
	14	66	m	cT4N0M0-cIIB	45	Local reprogression	13	ycT1cN0M0-ycIA3	IA3	Lobectomy	Combined SCLC + ADK	ypT1bNxM0-ypIA2	59 FOD
	15	72	m	cT1cN1M0-cIIB	54	Local reprogression	24	ycT2aN0M0-ycIB	IB	Lobectomy	SCLC	ypT1bN1M0-ypIIB	21 FOD
	16	70	m	cT3N2M0-cIIIB	45	Local reprogression	17	ycT2aN0M0-ycIB	IB	Lobectomy	SqCC ***	ypT2aN2M0-ypIIIA	25 FOD
Pan, 2017 [11]	17	54	m	cT2N2M0-cIIIA	-	Progression under CT	-	cT2N2M0-cIIIA	IIIA	Right pneumonectomy	SCLC	ypT2N2M0-pIIIA	>24 **** FOD

Abbreviations (alphabetical order): ADK = adenocarcinoma, CT = chemotherapy, DOD = died of disease, FOD = free of disease (free of malignancy), LCLC = large-cell lung cancer, LND = lymph node dissection, mo = months, RT = radiotherapy, SCLC = small-cell lung cancer, SqCC = squamous-cell carcinoma, we = weeks. Note: The presented patients received chemoradiation before the (un)planned salvation surgery (6 cases); one patient was planned to receive chemoradiation but underwent resection because of the progression under chemotherapy (case number 17). * The authors indicated 5 years, we converted this into 60 months for the relevance of data communication. ** All resections were accompanied by lymph node dissection, except for cases 12 and 13 (only wedge lung resections), as the authors described. *** “considered to be a part of combined SCLC”, as described by the authors. **** The authors described >2 years, we converted in >24 months for the relevance of data communication.

## 4. Discussion

### 4.1. Definitions of Salvage Surgery in Lung Cancer

Salvage surgery for primary lung cancer has started to be performed more recently compared to advanced colon cancer [13], esophageal cancer [14], malignant thymoma [15], or cervical cancer [16]. The efficacy and safety of salvage thoracic operations have not yet been fully elucidated [17]. The present definition of salvage surgery for lung cancer is younger than 20 years and refers almost exclusively to non-small cell lung cancer (NSCLC) [17,18,19,20,21,22,23,24,25,26,27,28,29,30], but it is important to underline that the same definition was proposed for and the same procedures were performed in SCLC, as presented above; therefore, the theoretical and practical information is resumed as follows:(a)Salvage surgery for an emergent complication, as a life-saving procedure, performed for an event that occurred during the natural history of the tumor or as a complication during oncological treatment, including SBRT, with curative or palliative intent: massive hemoptysis, lung abscess, empyema, broncho–pleural fistula;(b)Salvage surgery after definitive (full-dose) chemo–radiation therapy/after previous local (SABR) or general treatment (ex. targeted therapy): residual/persistent localized disease, relapsed tumor/recurrence after complete response, cases judged to be contraindicated for chemotherapy or definite radiation therapy due to severe comorbidities, despite a clinical diagnosis of NSCLC stage IIIA, IIIB, or IV disease initially, delayed decision to convert to a trimodal approach;(c)Salvage surgery for progression under chemotherapy;(d)Salvage surgery for oligo-metastatic disease.

We must clearly differentiate salvage surgery performed for early-staged tumors inadequately responsive to SBRT or targeted therapy from salvage surgery performed for locally advanced tumors, previously chemo- and/or radio-treated.

### 4.2. Salvage Surgery after a Full Dose of Chemo–Radiotherapy

Salvation surgery a after full dose of chemo-radiotherapy is usually technically difficult, the dissection of the pulmonary vessels and mediastinal lymph nodes being technically demanding; at least the same surgical technical difficulties are encountered after immunotherapy. Intrapericardial dissection may be necessary. Bronchial stump (or suture) needs protection and a healing adjuvant by covering with a well-vascularized flap. Complications are reported in 25–70% of the patients, depending on the careful selection and fitting of the patients. Major complications reported are pneumonia, arrhythmia, intrathoracic hematoma, major vessel injury, infarction/ventricular fibrillation arrest, recurrent laryngeal nerve paralysis, broncho–pleural fistula, acute respiratory distress syndrome (ARDS), chylothorax, stroke, heart failure, heart herniation, empyema, infected wound, etc. [17,31,32,33,34,35].

In the literature, 90-day mortality rates have been cited (ex. 6.7% [34], 2.8% [32], 7.7%, 0–11.4% [35]).

### 4.3. Salvage Surgery for SCLC

For patients with early-stage SCLC, the standard treatment recommended by guidelines is combined chemotherapy and concomitant radiotherapy [36]. Despite the aggressive initial treatment, most of the patients present a locoregional relapse of the disease or distant metastasis in the next 2 years [6,37,38]. For recrudescent SCLC, different modalities of treatment are described: second-line platinum-based chemotherapy, monotherapy or combination therapy, nivolumab as second-line therapy, and salvation surgery [6,36,38,39,40,41]. For each patient, the best option is proposed after a careful and complete evaluation of relapsed disease extent and medical fitness.

Salvage surgery for SCLC is performed, as we found in the literature, as a curative-intent lung resection:for tumor evolution under chemotherapy,after chemo-radiotherapy for residual lesions,after chemo–radiotherapy for local re-progression in the previously irradiated area [8,21].

Joosten and collab. stated in 2021 that, when choosing salvation surgery for SCLC, a multidisciplinary tumor board must evaluate the local extension to be technically resectable and if “the risks of salvage surgery outweigh the expected outcome with second-line chemotherapy” [6], as Anraku and Waddell discussed in 2006 [5].

Staging of the mediastinum before resection is recommended in all publications about salvation surgery. There is a need for disruptions in situations in which needle techniques (ex. EBUS-TBNA) are not available for restaging (and even for first staging) the mediastinum. In those situations, PET-CT may remain the only option for mediastinal staging, considering the risk of (re-)mediastinoscopy after definitive mediastinal radiotherapy or after another mediastinoscopy. By a careful individual evaluation of each patient and locally adjusted therapeutic protocols, the best option for every difficult case can be offered, including salvation surgery.

The data from the literature suggest that N2 disease is not an absolute contraindication for salvage surgery, if N2 can be completely resected with acceptable risk (but all cases were NSCLC) [7]. At this moment, there is only one case of salvage resection for N2 SCLC: the case published by Pan and co. in 2017 (Table 1, case 17). The indication for resection was made under the pressure of a growing tumor/relapse under chemotherapy. Time will demonstrate the feasibility of this management in N2 SCLC cases and the importance of a good clinical judgment of each case. New systemic personalized therapies will play an important role in these SCLC patients.

Of the 17 cases with salvage surgery for SCLC, 15 underwent an anatomic lung resection, and lymph node dissection was added for oncological radicality; in two cases, a wedge lung resection was performed to maximally preserve the lung parenchyma, and no therapeutic approach for the lymph nodes was performed in those two patients.

In 13 cases of anatomic lung salvage resection, the authors [6,8] affirmed the coverage of the bronchial stump with an intercostal muscle flap in 10 cases, and with a serratus anterior muscle flap, a pericardium flap, and an omentum flap, each in one case. The other authors did not specify if the bronchial stump was covered after resection.

Regarding postoperative complications after salvation surgery for SCLC, among the 17 cases from the literature, for 2 patients information was not available in the papers [7,11], and for the rest of the 15 patients, the authors published intrahospital complications: Joosten and collab. presented intrahospital complications in 5 patients, consisting of atrial fibrillation (2 cases), atrial fibrillation and pneumonia (1 case), air leakage (1 case), and respiratory distress (1 case) [6]; Nakanishi and collab. presented no complications in the 5 cases they operated [8]. We can conclude that postoperative morbidity after salvage surgery for SCLC in the published cases was 33.3% (5 patients from 15).

No intrahospital mortality was registered after salvage surgery for SCLC in the published papers.

The published survival of the patients subjected to salvation surgery for SCLC was between 2 and 114 months. At the publishing time, only four patients were deceased because of the disease (DOD,) and for one patient survival was not noted, the remaining 12 patients being alive and free of disease (FOD). The estimated median overall survival (OS) for this selected lot of patients was 86 months. The median estimated 2-year survival was 92%. The median estimated 3-year survival was 80%, and the median estimated 5-year survival was 66%. Analyzing those results, we observed a mean FOD survival of 3 years for patients who underwent salvation surgery performed for SCLC.

The different and unknown biological behavior of certain tumors and bio-immunological differences between patients may explain the out-of-statistics survival for the patients with salvation surgery performed for SCLC.

The major limitation of this review consists in the very small number of papers published on the topic, with only 17 cases presented in the literature regarding salvation surgery for SCLC after non-surgical definitive treatment.

## 5. Conclusions

The present indications for salvation surgery in SCLC are: evolution under chemotherapy; residual lesion after chemo–radiotherapy; local re-progression in the previously irradiated area after chemo–radiotherapy. In highly selected SCLC patients after chem–radiation therapy (fit for surgery, M0, expected R0, preferably non-N2, and preferably with confirmed viable tumor), salvage surgery should be discussed within an experienced multidisciplinary lung cancer team. In these patients, the desirable long-term survival may be possible. Salvage lung resection for small-cell lung cancer (SCLC) may be a reasonable treatment in very highly selected patients, as an alternative to second-line chemotherapy, offering a good local control and a favorable survival outcome, but clearly, this should be demonstrated in clinical trials.

Salvage lung resection after definitive chemoradiation is feasible, with acceptable postoperative survival and complication rates, in SCLC as in NSCLC, and perioperative mortality appears acceptable in NSCLC, though not yet published in SCLC, because this is a very rarely accepted indication, even in high-volume surgery centers. At present, only retrospective, very small series or isolated cases are available. Carefully designed prospective studies are necessary to define more precisely the indications and results of salvage surgery following full-dose chemoradiation therapy for locoregionally advanced disease in SCLC, but also in NSCLC patients.

## Figures and Tables

**Figure 1 cancers-15-02241-f001:**
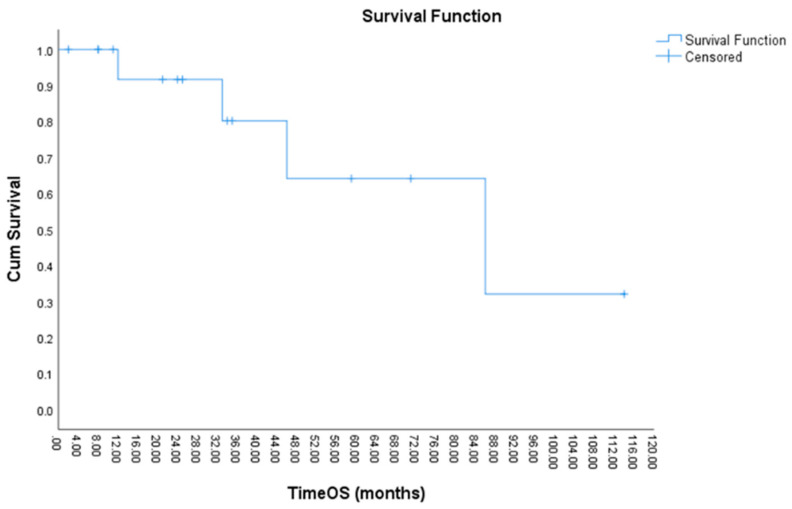
Estimated median overall survival (OS) of patients with SCLC receiving salvage surgery, using the Kaplan–Meier method.

## Data Availability

Not applicable.

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
