# Peer review of "Salvage Surgery for Small-Cell Lung Cancer—A Literature Review"

_cancers, 2023, doi:10.3390/cancers15082241_

Round 1

Reviewer 1 Report

1, Line 123  word error : The time between edn of radiation. 

2, Line 226,  Evolution means progression in sentence?: Salvage surgery for evolution under chemotherapy.

3, Line 313, “In highly selected SCLC patients”is recommended to be revised to “in highly selected SCLC patients after chemoradiation”. 

4. In the section of conclusion, some limitations of this paper shoulder be added such as : small samples and case selection bias. There are other poor survival data about salvage surgery in similar SCLC patients, which tend to be neglected if unpublished.

Author Response

Reviewer 1:

Answers to Revier 1:

Thank you for your careful reading of our paper. We are happy to respond to your comments and clarify the pointed aspects.

1, Line 123  word error: The time between edn of radiation.  – corrected, thank you!

2, Line 226,  Evolution means progression in sentence?: Salvage surgery for evolution under chemotherapy. – corrected “progression” , thank you!

3, Line 313, “In highly selected SCLC patients”is recommended to be revised to “in highly selected SCLC patients after chemoradiation”. - corrected, thank you!

  1. In the section of conclusion, some limitations of this paper shoulder be added such as : small samples and case selection bias. There are other poor survival data about salvage surgery in similar SCLC patients, which tend to be neglected if unpublished. – we added the following phrase at the end of Discussions “”The major limitation of this review consists in the very small number of papers published on the topic, with only 17 cases presented in the literature regarding salvation surgery for SCLC after non-surgical definitive treatment.”” . Thank you!

We highly appreciated all the suggestions as we think they enriched our article. Thank you!

Reviewer 2 Report

Dear Editor and Authors,

Thank you very much for asking me to review and evaluate this manuscript titled “Salvage Surgery for Small-Cell Lung Cancer - Literature review” by Dr. Motas and colleagues from the “Carol Davila” University of Medicine and Pharmacy in Bucharest, Romania.

In this review the authors try to provide an overview of a quite interesting subject, that of salvage surgery for Small-Cell Lung Cancer which is quite rare, not widely applicable and as such there are very limited information available. Therefore, a comprehensive review of available cases can be valuable to the clinical/surgical community.

The authors present a total of 17 patients utilizing reported case reports, compile their data and attempt to conduct a basic analysis!

The manuscript needs some significant language correction because it is challenging at certain places to understand what the authors are trying to convey. Therefore, there are a number of issues which I feel need addressing. Specifically:

1. In terms of the search terms the authors do not seem to completely list the terms used - they do not seem to list “lung cancer”, “small cell lung cancer” as searchable terms!

2. The authors report that papers not written in English were also reported. In which additional languages were they able to find cases in? Can they list them?

3. How did surgery not included in the initial management plan be performed with an intent to cure? Can the authors please explain this? Lines 79 - 80 need some clarification!

4. “Cases with salvage surgery for NSCLC” were excluded from the search!! Shouldn’t the breakdown occur much earlier on i.e. SCLC vs. NSCLC histological diagnosis?? Again, this section is not clear!

5. The number of patients “analyzed” is not surprising quite small at 17 cases. What type of statistical analysis did the patients perform (using SPSS) for so few cases! This is basically a case series presentation which does not even include averages!!

6. Table 1 needs reworking! Specifically, units such as days or months are needed for the category “Time from RT to surgery”. Also, it would be better-easier to report the actual stage of the patients (IIIB, IV) and not only the TNM! The pre’ and post therapy stage needs to be reported to compare response!

7. The indication for surgery in most cases was recurrence of disease!! How does this fit with the concept of downstaging mentioned before and the concept of “curative salvage surgery”?

8. I suggest to the authors that Table 1 become a supplemental table or table 2 and instead construct a concise, comprehensive Table 1 were variable averages are calculated and reported i.e. average age, percentage male vs female, stage % - II/III/IV ect!

9. What do the authors mean when they report in 2 cases (Joosten, 2021) that no malignancy was found?? Does it mean there was no cancer at all? Did these 2 patients downstage and their cancer go away and how does this correspond with recurrence and persistent disease which are reported as the reasons for their surgery?? This is confusing and needs clarification please.

10.  When reporting the survival, it is better to utilize the same variable/measurement! Cant use years in some cases and months in others!! Nor can you use more than 8 or more than 2 years ect. Use exact times and in the same measurement/unit i.e. use months in all!! How was the Kaplan Meir survival plot was constructed without exact data??

11. Apart from the radiotherapy dose were the authors able to mine other data regarding the patients such as type of radiotherapy used or chemotherapy regiment(s) used? Or other demographic or treatment data?

12. The discussion gives a broad overview of the general background regarding salvage surgery and management of SCLC but do not tie well the new information from the conducted analysis!! No mention is made on the benefits shown in these patients due to salvage surgery (if any) and how their survival compared to the average!! At the end was there a benefit for doing salvage surgery? Can the authors discuss this as a main point!

In conclusion, as previously mentioned this pooled analysis can be useful in an area with quite sparce information available, however it needs some major improvement/re-analysis to make it more understandable and useful to the reader! Thank you again for asking me to review this work. I am awaiting the authors to do their best! Kind regards to all. 

Author Response

Reviewer 2:

Answers to Reviewer 2:

Thank you for your careful reading of our paper. We are happy to respond to your comments and clarify the pointed aspects.

Dear Editor and Authors,

Thank you very much for asking me to review and evaluate this manuscript titled “Salvage Surgery for Small-Cell Lung Cancer - Literature review” by Dr. Motas and colleagues from the “Carol Davila” University of Medicine and Pharmacy in Bucharest, Romania.

In this review the authors try to provide an overview of a quite interesting subject, that of salvage surgery for Small-Cell Lung Cancer which is quite rare, not widely applicable and as such there are very limited information available. Therefore, a comprehensive review of available cases can be valuable to the clinical/surgical community.

The authors present a total of 17 patients utilizing reported case reports, compile their data and attempt to conduct a basic analysis!

The manuscript needs some significant language correction because it is challenging at certain places to understand what the authors are trying to convey. Therefore, there are a number of issues which I feel need addressing. Specifically:

  1. In terms of the search terms the authors do not seem to completely list the terms used - they do not seem to list “lung cancer”, “small cell lung cancer” as searchable terms! – at submission we provided the following keywords: “”small-cell lung cancer SCLC; salvage surgery; salvage resection; lung resection; definitive chemoradiotherapy; locally advanced lung cancer; definitive surgery””, which covers your suggestions. Nevertheless, we added the simple lung cancer as well and salvation surgery and resection after chemoradiation. Thank you!
  2. The authors report that papers not written in English were also reported. In which additional languages were they able to find cases in? Can they list them? – We decided to eliminate this sentence in order to avoid further misunderstandings. Thank you!
  3. How did surgery not included in the initial management plan be performed with an intent to cure? Can the authors please explain this? Lines 79 - 80 need some clarification! - Dear Reviewer 2, for clear presentation of the surgical indications in SCLC, we added in Introduction a general clarification: “SCLC is characterized by rapid growth and early lymphatic and hematogenous metastases – for this reason, the surgical resection is not the first therapeutic option; in fact, surgery is not an option for most of the patients with SCLC “. So, the lines 79-80 became clear: the selected patients followed a complete non-surgical treatment (because surgery is not indicated and not possible), after which, for the relapsed tumor, salvage surgery was possible to be performed as an exceptional treatment, with clear benefits for these highly selected patients. After this addition, the second paragraph of the Introduction becomes clearer: “Salvage surgery is a relatively new entity in thoracic surgery and oncology. In the context of modern standard therapies for SCLC, we found the first indications for salvage surgery in SCLC in a publication from 2006 (Anraku and Waddell): chemo-resistant localized SCLC, or local relapse after an initial response to chemo/chemoradiotherapy, for the reason that resection may be more effective than second-line chemotherapy [5]”. An explanation for our readers not familiarised with SCL, we presented the paragraph in Discussions, lines 284-291: “For patients with early-stage SCLC, the standard treatment recommended by guidelines is combined chemotherapy and concomitant radiotherapy [36]. Despite aggressive initial treatment, most of the patients present locoregional relapse of the dis-ease or distant metastasis in the next 2 years [6,37,38]. For recrudescent SCLC, differ-ent modalities of treatment are described: second-line chemotherapy platinum-based, monotherapy or combination, nivolumab as second-line, and salvation surgery [6,36,38,39-41]. For each patient the best option is proposed after a careful and complete evaluation of relapsed disease extent and medical fitness”. Thank you!
  4. “Cases with salvage surgery for NSCLC” were excluded from the search!! Shouldn’t the breakdown occur much earlier on i.e. SCLC vs. NSCLC histological diagnosis?? Again, this section is not clear! – Because the search methodology included the salvation surgery, the bulky returned as NSCLC – which were, obviously, excluded, and only SCLC were kept. The SCLC (small-cell lung cancer) is about 6-10 times rarer then NSCLC (non small-cell lung cancer), and the articles about these cancers SCLC vs NSCLC keep the parity. Further in the paper, you can find comparison between results and outcome after salvage surgery for NSCLC and SCLC, because both components of lung cancer are compared in the literature and in daily practice. Thank you!
  5. The number of patients “analyzed” is not surprising quite small at 17 cases. What type of statistical analysis did the patients perform (using SPSS) for so few cases! This is basically a case series presentation which does not even include averages!! – We have to present the SPSS in Matherials and Method, as part of the template, even if we use it for just one function – as presented in Figure 1. Thank you!
  6. Table 1 needs reworking! Specifically, units such as days or months are needed for the category “Time from RT to surgery” – added, thank you!. Also we converted the 2 years in 24 months at Survival for the patient number 17. Also, it would be better-easier to report the actual stage of the patients (IIIB, IV) and not only the TNM! – We kept the reported data, as presented by the authors of the articles (references 6, 7, 8 and 11). So it was more clear to present for our readers the TNM details because it offers info about tumor dimensions and lymph nodes invasion. The pre’ and post therapy stage needs to be reported to compare response! – We added a column in Table 1 containing the stage before surgery cTNM. Thank you!
  7. The indication for surgery in most cases was recurrence of disease!! How does this fit with the concept of downstaging mentioned before and the concept of “curative salvage surgery”? - The presented SCLC patients were treated by chemo-radiation with definitive curative intent, and NOT for downstaging (this being a specific treatment for stage III NSCLC and NOT for SCLC). After a while, the recurrence occurred. The patient was reevaluated and the salvation surgery (with curative intention) was performed as a part of the management of the recurrence. Thank you!
  8. I suggest to the authors that Table 1 become a supplemental table or table 2 and instead construct a concise, comprehensive Table 1 were variable averages are calculated and reported i.e. average age, percentage male vs female, stage % - II/III/IV ect! – We did our best to gather the data from the literature (please see references 6, 7, 8 and 11) and to uniformize their presentation. Thank you!
  9. What do the authors mean when they report in 2 cases (Joosten, 2021) that no malignancy was found?? Does it mean there was no cancer at all? Did these 2 patients downstage and their cancer go away and how does this correspond with recurrence and persistent disease which are reported as the reasons for their surgery?? This is confusing and needs clarification please. – As we all know, after oncological treatment, the viable tumor may completely disappear and the remaining scar tissue with inflammation uptakes the FDG and gives false-positive results on PET-CT. For this reason, an imagistic relapse is excised and the pathologist colleagues find no viable tumor, only chronic inflammation and fibrotic scar tissue. We see this more frequently in NSCLC, but it appears also in SCLC. In fact, it may appear in cancers developing in all organs, and the best example from our practice, besides lung cancer, are mediastinal lymphomas. Thank you!
  10.  When reporting the survival, it is better to utilize the same variable/measurement! Cant use years in some cases and months in others!! Nor can you use more than 8 or more than 2 years ect. Use exact times and in the same measurement/unit i.e. use months in all!! How was the Kaplan Meir survival plot was constructed without exact data?? – Please see previous corrections, thank you. Also, we add that those are the data provided in the literature, which we reviewed and corroborated in order to drown pertinent conclusions. Thank you!
  11. Apart from the radiotherapy dose were the authors able to mine other data regarding the patients such as type of radiotherapy used or chemotherapy regiment(s) used? Or other demographic or treatment data? – As previously presented, those are the data existing in the literature. Thank you!
  12. The discussion gives a broad overview of the general background regarding salvage surgery and management of SCLC but do not tie well the new information from the conducted analysis!! No mention is made on the benefits shown in these patients due to salvage surgery (if any) and how their survival compared to the average!! At the end was there a benefit for doing salvage surgery? Can the authors discuss this as a main point! – Please see the lines 271-271 from Discussions: “Joosten and colab. state in 2021 that, when choosing the salvation surgery for SCLC, the multidisciplinary tumor board must evaluate the local extension in order to be technically resectable, and “the risks of salvage surgery outweigh the expected out-come with second-line chemotherapy” [6], as Anraku and Waddell discussed back in 2006 [5].” Thank you!

In conclusion, as previously mentioned this pooled analysis can be useful in an area with quite sparce information available, however it needs some major improvement/re-analysis to make it more understandable and useful to the reader! Thank you again for asking me to review this work. I am awaiting the authors to do their best! Kind regards to all. - We highly appreciated all the suggestions as we think they enriched our article. Thank you very much!

Round 2

Reviewer 2 Report

Dear Authors,

I have re-read and re-evaluated the modified manuscript and I have taken into account your responses to my commentary. Thank you for your effort. I am now content to recommend the publication of your work.

Kind regards,